# Polydopamine Incorporation Enhances Cell Differentiation and Antibacterial Properties of 3D-Printed Guanosine-Borate Hydrogels for Functional Tissue Regeneration

**DOI:** 10.3390/ijms24044224

**Published:** 2023-02-20

**Authors:** Maria Merino-Gómez, Javier Gil, Roman A. Perez, Maria Godoy-Gallardo

**Affiliations:** 1Bioengineering Institute of Technology (BIT), International University of Catalonia (UIC), Carrer de Josep Trueta, Sant Cugat del Vallès, 08195 Barcelona, Spain; 2Department of Dentistry, Faculty of Dentistry, International University of Catalonia (UIC), Carrer de Josep Trueta, Sant Cugat del Vallès, 08195 Barcelona, Spain

**Keywords:** nucleoside-based hydrogels, guanosine and derivatives, 3D printing, polydopamine, antibacterial activity

## Abstract

Tissue engineering focuses on the development of materials as biosubstitutes that can be used to regenerate, repair, or replace damaged tissues. Alongside this, 3D printing has emerged as a promising technique for producing implants tailored to specific defects, which in turn increased the demand for new inks and bioinks. Especially supramolecular hydrogels based on nucleosides such as guanosine have gained increasing attention due to their biocompatibility, good mechanical characteristics, tunable and reversible properties, and intrinsic self-healing capabilities. However, most existing formulations exhibit insufficient stability, biological activity, or printability. To address these limitations, we incorporated polydopamine (PDA) into guanosine-borate (GB) hydrogels and developed a PGB hydrogel with maximal PDA incorporation and good thixotropic and printability qualities. The resulting PGB hydrogels exhibited a well-defined nanofibrillar network, and we found that PDA incorporation increased the hydrogel’s osteogenic activity while having no negative effect on mammalian cell survival or migration. In contrast, antimicrobial activity was observed against the Gram-positive bacteria *Staphylococcus aureus* and *Staphylococcus epidermidis*. Thus, our findings suggest that our PGB hydrogel represents a significantly improved candidate as a 3D-printed scaffold capable of sustaining living cells, which may be further functionalized by incorporating other bioactive molecules for enhanced tissue integration.

## 1. Introduction

Supramolecular hydrogels have gained a lot of interest in the past three decades due to their similar water content and elasticity to natural tissue, and thus they hold great promise in a variety of applications [1,2,3]. While their ability to absorb large amounts of water results from hydrophilic groups in the building blocks, physical (i.e., non-covalent) interactions and crosslinks between the formed chains ensure their resistance to dissolution. Importantly, supramolecular hydrogels are considered a novel class of soft materials that are crosslinked by non-covalent interactions (e.g., hydrogen bonds, ionic salt bridges, metal-to-ligand interactions, π–π interactions). The high association and dissociation rate of such non-covalent interactions endows supramolecular hydrogels with important dynamic behaviors. Therefore, supramolecular hydrogels have the ability to remodel, reshape, and adapt to environmental conditions and changes. These dynamic features, such as stimuli responsiveness [4,5], self-healing [6,7,8,9], and shear thinning [10], are crucial for a variety of emerging applications including therapeutic delivery [11,12,13,14], tissue engineering [15,16], and bioimaging [17].

Considering the binding motif, supramolecular hydrogels can be categorized as either polymeric supramolecular, molecular, or hybrid supramolecular. Polymeric supramolecular hydrogels are created by the physical crosslinking of polymers modified with so-called supramolecular motifs. Molecular hydrogels, on the other hand, result from the self-assembly of low-molecular-weight (LMW) building blocks, forming first nuclei and nanofibers and ultimately a complex three-dimensional (3D) network through crosslinking and entanglement. Finally, in hybrid supramolecular hydrogels, predefined nano-structures are incorporated to provide additional multivalent crosslinking sites.

As building blocks for molecular hydrogels, several natural compounds such as amino acids, peptides, fatty acids, sugars, and nucleobases (e.g., adenine, cytosine, guanine), nucleosides (i.e., a nucleobase with a five-carbon sugar; e.g., adenosine, cytidine, guanosine (Guo)), and nucleotides (i.e., a nucleoside with one, two, or three phosphates attached; e.g., adenosine mono-, di-, and triphosphate) have been used [18].

In previous reports, various nucleobases have been functionalized and derivatized and their gelation properties characterized [18]. Guo represents a natural nucleoside and an important LMW building block for supramolecular hydrogels due to its intrinsic self-assembly properties.

In particular, compounds containing the nucleobase guanine frequently undergo quadruple association into G-quartets via Hoogsteen-type hydrogen bonding, thereby forming supramolecular systems that stack into G-quadruplex (G4) assemblies in the presence of small cations such as potassium (K^+^). Under certain conditions, a complex nanofilament network capable of encapsulating enough water to produce a hydrogel can be formed (Figure 1).

The last years have experienced an increasing interest in Guo-based hydrogels due to the good biological compatibility of the G4 superstructures, and many studies have focused on developing innovative Guo-based hydrogels with enhanced lifetime stability [18,19,20]. However, only a few successful systems have been reported to date, and their usage as (bio)inks is in particular constrained due to their persistent stability issues [12,21,22]. Several strategies have focused on changing the mechanical and chemical structure of the Guo-based hydrogels, either by using new Guo derivatives [18] or by developing binary hydrogel formulations [22]. For example, an improved lifetime stability was observed for binary hydrogels prepared from mixtures of Guo with guanosine 5′-monophosphate (GMP) [23,24,25] or isoguanosine (isoGuo) [26].

Notably, this improved both their printability and stability. Additionally, Peters et al. [20] described that the addition of K^+^ and borate anions into Guo solutions was crucial for the controlled gelation and improved stability of the formed so-called Guo-boric acid (BA) (GB) hydrogels.

Extensive research has been conducted on how to modify various biomaterials to enhance their biological responsiveness. For example, mussel-inspired polydopamine (PDA) has generated a lot of interest as a novel method for biomaterial functionalization because of its simple preparation, good biocompatibility, material independence in regard to deposition, reported favorable interactions with living cells, and high chemical reactivity for additional functionalization [27,28,29]. Compared to other materials used for surface functionalization, the main advantages of PDA are that (i) synthesis is relatively easy and no organic components are needed, (ii) particle size and film thickness can be easily controlled by simple parameters such as pH, temperature, and dopamine (DA) concentration, and (iii) the catechol/quinone moieties of PDA offer interaction points for both physical (e.g., via π-π stacking or hydrogen bonding) and chemical (e.g., via Michael addition or Schiff base reactions) bonding. Crucially, it has also been shown that PDA enhances cell attachment, proliferation, and differentiation of, for example, endothelial [30] and bone marrow stromal cells [31].

Microbial infections represent a major public health problem, particularly when they are associated with biomaterial implantation (biomaterial-associated infection; BAI) [32]. Such infections are typically caused by 3–10 dominant microorganisms, including *Staphylococcus aureus* (*S. aureus*), *Staphylococcus epidermidis* (*S. epidermidis*), and *Escherichia coli* (*E. coli*) [33,34,35,36]. Notably, the major pathogen *S. aureus* is the most frequent species detected in BAIs, accounting for ~70% of wound-colonizing bacteria. Thus, there is an urgent need to improve the antibacterial properties of biomaterials and, ideally, to develop strategies capable of overcoming antibiotic-resistant bacteria. As a functional but insoluble biomaterial, PDA has been used for antibacterial applications not only as a functionalization layer to immobilize antibacterial agents on biomaterial surfaces but also as a bactericidal compound in itself [37,38]. The main antibacterial mechanisms and pathways may be divided into four categories: (i) Contact active antibacterial effect, i.e., PDA can destroy the structure of the bacterial cell membrane by direct interaction with surface-located proteins by metal chelation or electrostatic interactions. (ii) Photothermal conversion capacity, i.e., PDA’s conjugated electron donor–acceptor structures indole-5,6-quinone and 5,6-dihydroxyindole can synergistically absorb a wide range of light, leading to localized heat generation and thus damage of adjacent bacterial cells. (iii) Generation of reactive oxygen species (ROS), i.e., caused by the electron transfers during phenolic quinone isomerization and ion chelation by the catechol moieties of PDA, ROS are generated in close proximity to the bacterial cell surface, thus damaging bacterial cell wall integrity. (iv) Directed chemical modification of PDA, for example, by converting PDA to N-halamine by chlorination, which increases the antibacterial properties of PDA owing to the added destructive power of halogen ions [39]. All of these factors have sparked intense interest in developing new antibacterial treatment strategies based on PDA. In particular, we believe that trapping PDA inside hydrogel matrices might represent a highly effective augmentation for improving the antibacterial properties of the biomaterial and thus improve its suitability for soft tissue repair.

Hydrogel-based biomaterials, such as alginate-based hydrogels [40], gelatin hydrogel microspheres [41], and nanofiber-reinforced chitosan hydrogels [40] have been described for tissue regeneration. However, for successful application, such hydrogels need to possess the ability to be formed under physiological conditions and maintain or recover their original form and viscosity when subjected to stress [9]. Notably, this reassembly capacity is equally essential for the 3D printing of hydrogels, an effective approach for creating customized scaffolds with controlled macro- and microporous structures [36]. Moreover, such hydrogels may then be directly injected at defect sites, further expanding their applicability. Thus, we aimed to develop a versatile polydopamine-guanosine-borate (PGB) hydrogel for tissue engineering. While simple Guo-based hydrogels typically exhibit adverse material spreading after printing and lack the often-required long-term stability, we proposed that utilizing Guo, BA, and DA will result in hydrogels with ideal thixotropic properties while also enhancing life-time stability, in particular due to the outstanding adhesive capacity of PDA and its ability to form coatings on all kinds of materials. Moreover, PDA is described to promote cell proliferation and osteogenic differentiation; thus, PGB hydrogels may be more effective for tissue regeneration than traditional Guo-based hydrogels. In addition, its reported antibacterial properties may reduce the risk of biomaterial-associated infections (BAIs) and thus implant failure, another unique feature. Notably, Guo-based hydrogels are also suitable for drug delivery applications as they allow for the non-covalent encapsulation of bioactive molecules into their nanofibrillar hydrogel network, resulting in a steady release at the defect site, and thus may improve tissue regeneration. In addition, as PDA was shown to enhance the loading and release capability of hydrogels [41], we consider the herein-developed PGB hydrogel as a prime candidate for tissue reconstruction and drug delivery applications.

To evaluate the potential of PGB hydrogels for tissue regeneration, we (i) synthesized various PGB hydrogels with different dopamine concentrations and evaluated their 3D printability, and (ii) we assessed their stability and diffusion properties. Furthermore, (iii) various mechanical and viscosity properties of the different PGB hydrogels were analyzed, (iv) antibacterial properties against bacterial adhesion and early biofilm formation were evaluated, and finally, (v) rat mesenchymal stem cells (rMSCs) were seeded on top of the most promising hydrogel composition to investigate cell cytotoxicity, migration, and osteogenic potential.

## 2. Results and Discussion

Many recent studies have concentrated on the development of inks based on Guo hydrogels due to their nanofibrous network [23,24,25,42,43]. Their strong fibrous matrix allows for physical scaffolding within the extracellular matrix (ECM) via non-covalent, intermolecular interactions, and their high water-holding capacity provides an environment suitable for cell survival and differentiation [20]. Furthermore, Guo hydrogels can be injected into specific target locations due to their dynamic behavior and allow cells to migrate and propagate within the scaffold. However, this dynamic nature decreases their mechanical strength and long-term stability. In this project, we combined Guo-based hydrogels with PDA [27,28,44,45], a bioinspired synthetic polymer, to optimize the balance between hydrogel stability and dynamicity. PDA recently emerged as a powerful biomedical tool for various applications as it exhibits great adhesive properties for a variety of surfaces (e.g., metallic, polymeric, ceramic) [46,47], improves their fiber/matrix interfacial adhesion, and can be applied as continuous layers with simplicity [48,49].

### 2.1. Optimization and Assessment of PGB Hydrogel Formation

Over recent years, several studies confirmed that GB hydrogels have better stability and longevity than Guo-only hydrogel formulations [19,20], and hence GB hydrogels are now regarded as the gold standard of Guo-based hydrogels [18,19,50,51]. Following a recently developed protocol for self-assembled injectable GB hydrogels (to be published), a GB hydrogel solution (90 mM Guo, 40 mM BA, and 40 mM potassium hydroxide (KOH)) was mixed with DA at various concentrations (2.5, 5.0, 7.5, 10, and 12.5 mM). To ensure maximal PDA loading into the hydrogel, we measured the fluorescence intensity reading of formed PDA after distinct incubation times and using different DA concentrations. Importantly, after successful polymerization into PDA, DA exhibits a distinctive emission peak at 310 nm at an excitation wavelength of 260 nm (Figure 1a; control samples are shown in Appendix A). Simultaneously, PDA formation leads to a deep brown coloring of the hydrogel solution (Figure 1a, right inset). We detected a time-dependent and linear increase in fluorescence intensity for all tested DA concentrations, and PDA formation appeared complete after 5 h in all conditions (Figure 1(bi)). Notably, once the hydrogels were printed and immersed in a complete medium, the supernatants of PGB hydrogels prepared with 10 and 12.5 mM DA turned deep brown and opaque, while the ones prepared with 2.5, 5.0, and 7.5 mM stayed translucent (Figure 1(bii)). Thus, we concluded that both 10 and 12.5 mM DA were too much to be accommodated by the hydrogel, leading to the polymerization of considerable amounts outside in the surrounding fluid. Thus, we continued the optimization of gel-embedded PDA using DA concentrations of 2.5, 5.0, and 7.5 mM. Of note, others previously reported that BA/DA ratios of higher than 3 disallow the auto-oxidation of DA into PDA [52]. However, in our situation, most of the BA is already involved in the guanosine borate hydrogel network and is thus not able to interfere anymore with PDA formation.

Next, we assessed the precise amounts of PDA formed and immobilized inside the GB hydrogels by quantifying the fluorescence intensity of the corresponding PGB hydrogel. The results show that 0.13 ± 0.01, 0.12 ± 0.01, and 0.12 ± 0.01 mg mL^−1^ of PDA were effectively immobilized for 2.5, 5.0, and 7.5 mM of DA in the formulation, respectively. To further evaluate the efficacy of the PDA formation, the same reaction was performed in parallel in solution. Notably, while approximately 40 to 50% of the DA was converted into PDA in solution (49, 43, and 39% for 2.5, 5.0, and 10 mM DA, respectively), these rates were dramatically reduced in the hydrogels with 5 and 10 mM DA (16 and 11%, respectively), while for 2.5 mM DA, 73% of the DA polymerized when compared to the same reaction in solution. This is an important observation since it shows that the maximum amount of PDA covalently attached to our developed hydrogel scaffold is ~0.13 mg mL^−1^ and that the amount is independent of additional monomeric DA added to the PGB hydrogel formulation (i.e., 2.5 mM of DA is sufficient to produce the maximum amount of PDA that the GB hydrogel is capable to absorb).

### 2.2. Semi-Quantification of Printability

The combination of 3D-printing technology with recent advances in hydrogel formulations enables the formation of predefined structures with patient-specific geometries at high detail and provides a favorable environment for cell growth and differentiation [53].

After determining the optimal DA concentrations, we used two different tests to identify the ideal hydrogel composition in terms of 3D printability. In particular, four distinct parameters [54,55,56] were evaluated to assess and score our hydrogel formulations (Figure 2). For details on the used platform and the printing pattern, please see the Materials and Methods section. The pore area of the printed pattern was measured using ImageJ software (National Institutes of Health and LOCI, v1.48, Bethesda, MD, USA, and Madison, WI, USA) [57], and all equations are described in Section 3 Materials and Methods “Semi-Quantification of Printability”.

The qualitative evaluation of the collapse test for the various PGB hydrogel formulations (Figure 2(ai)) revealed that the printed filament did not collapse or deform after 1 min of rest. Interestingly, when DA was employed vs. the control hydrogel without, the collapse area factor (C_f_) increased, although the differences were minor. A simple explanation might be the higher gravity caused by the incorporated PDA and therefore the increased density and weight of the hydrogel. However, the data show that PDA has no significant effect on the collapse stability of the system.

To determine the filament fusion and pore shape fidelity of each hydrogel formulation, the diffusion rate (D_fr_), printability score (P_r_), and angular deviation rate (D_a_) were determined (Figure 2(aii–aiv)). All experiments were carried out on a plastic surface with two layers of hydrogel printed. However, just the top layer was analyzed in order to minimize the influence of the plastic–hydrogel interphase. For D_fr_ and P_r_, no significant differences were observed across all tested PGB hydrogels. Importantly, when capillary forces surpass the yield strength of the printed soft material, corner deformation increases, which can ultimately result in pore closures [54]. Additionally, no deformation or fracturing of the filaments was observed during printing, indicating that the hydrogel viscosities were appropriate [58]. However, pores of modest dimensions (1 × 1 and 2 × 2 mm) were found closed due to a rather wide printer nozzle (0.58 mm) as well as minor filament spreading.

Next, to assess the ink behavior during changes in printing direction, the angle printing was evaluated. For this, the top right square of each printed pattern was used for determining the D_a_ by ImageJ [57]. In detail, the center points of each corner were joined and the enclosed angles were calculated. All of the tested inks had D_a_ values of less than 3%, with the hydrogel using 7.5 mM of D_a_ performing best with a D_a_ of just under 2%.

To guarantee a parameter-based semi-quantitative assessment of the printability qualities, a dedicated scoring scheme was developed. Figure 2(bi) summarizes the normalized scores ranging from 1 to 10 of all hydrogels tested, with compositions using 0 and 5 mM DA demonstrating the highest overall performance across all criteria (Figure 2(bii)). Importantly, the obtained results are rather objective and hence may be used to compare (bio)inks of widely different compositions. However, more parameters may be necessary to establish a meaningful comparison for extremely divergent (bio)ink families. As PGB hydrogels prepared with 5 mM DA showed overall the best hydrogel printing properties, subsequent experiments were performed using this hydrogel formulation.

### 2.3. PGB Hydrogel Characterization

To gain a better understanding of the PDA coating effect on GB hydrogels, we examined the stability of the printed PGB hydrogel by incubating it in a complete medium at 37 °C for up to 14 days. Over the first 48 h, a reduction of ~30% in the normalized scaffold area was observed. This might be explained by the contraction or shrinkage of the hydrogel due to water loss from the hydrogel network as a result of the higher ionic strength or a difference in pH in the surrounding medium (Figure 3a) [59]. However, no significant differences were seen between GB hydrogels with and without PDA coating. Thus, we next assessed the development of the pore area over time (Figure 3b), a key parameter of hydrogel stability, as water loss from the printed hydrogel, for example, due to a higher ionic strength or different pH in the surrounding medium, can lead to significant hydrogel contraction (shrinkage) [59]. After an initial ~20% decrease in the first 3 days, the pore area of the PGB hydrogel stabilized and did not decrease much more throughout the remaining days. On the other hand, the pore area of GB hydrogels dropped significantly and consistently over 14 days, indicating that the PDA coating of the network indeed led to a more stable hydrogel by increasing its robustness and resilience.

Cell migration into and also within printed 3D scaffolds is critical for successful tissue regeneration of considerable size defects, as otherwise, the diffusion limits of oxygen, nutrients, and metabolites in the scaffold may obstruct scaffold integration [60,61,62]. Thus, we performed a series of in vitro tests. After 10 h of immersion in three different fluorescein isothiocyanate-dextran (FITC-Dextran) solutions, ~50% of the 70, 500, and 2000 kDa molecules (~1.8 to 2.2 mg) were able to diffuse into 1.5 mL of printed hydrogel (Figure 4). Next, the FITC-loaded hydrogels were immersed in fresh media for the same time period, which led to the release of 25 to 45% of the previously entrapped molecules. Notably, the 70 kDa FITC-Dextran was retained best, while the 500 and 2000 kDa versions behaved very similarly. Next, we reloaded and washed the PGB hydrogels two times more, both times increasing the amount of adsorbed FITC-Dextran for all three molecular weight variants over time. In each step, the amount of loaded FITC-Dextran increased by 42 to 77%, resulting in a final loading capacity of 6.0, 4.4, and 4.9 mg of FITC-Dextran, respectively. For GB hydrogels, the same trend was observed but significantly less FITC-Dextran was retained in the network. For example, initial FITC-Dextran loading resulted in only a ~20% uptake of the provided molecules into the hydrogel, which was also nearly completely removed during the following wash step. Only subsequent loading/washing cycles enabled an entrapment of FITC-Dextran molecules, leading to a final loading capacity of 3.0, 2.1, and 3.3 mg of 70, 500, and 2000 kDa FITC-Dextran, which is approx. 45-30% less than in the PDA-treated ones. This demonstrates that the GB hydrogel network comprises FITC-Dextran interaction sites, which are amplified by the presence of PDA. This is consistent with previous reports stating that PDA coatings have a high reactivity for secondary functionalization due to their many functional groups and can likewise bind or retain a wide range of bioactive molecules, such as proteins, peptides, and growth factors, by lowering their diffusivity [63,64].

Moreover, as our primary aim was to develop an ink with increased cell viability properties and minimal cell death, we assessed the pH of both the GB and PGB hydrogel formulations. The PDA-loaded hydrogel showed a pH of 7.70 ± 0.06, while the GB hydrogel was 7.68 ± 0.03. Given that physiological pH ranges from 7.34 to 7.45 [65] and given the effect of pH on primary cell function and would healing [66], we may infer that the pH of both hydrogels is most likely acceptable for cell survival.

### 2.4. Rheology Studies of PGB Hydrogels

Next, the mechanical properties of the PGB hydrogels were investigated using rheological experiments. Both the elastic and fluidity properties of the PGB hydrogel were measured using the storage modulus (G′) and the loss modulus (G″), which were determined utilizing a strain sweep, dynamic step-strain, and a peak hold assay.

First, the elastic nature of the hydrogel was evaluated using strain amplitudes ranging from 0.01 to 100% at an oscillation frequency of 1 Hz (Figure 5(ai)). At low shear strain values, G′ was larger than G″. However, a turning point was identified at an oscillation strain of 1.24%, and beyond that point, the revere behavior was detected (1.26, see Appendix A). The observed trend is caused by the breakdown of intermolecular or π- π stacking interactions, as the system is no longer in the linear viscoelastic region (LVE). Thus, to ensure an elastic behavior of the PGB hydrogel, we used a strain of 0.1% in subsequent experiments [67,68].

Next, to study the self-healing potential and recovery of the PGB hydrogels, we performed a so-called recovery test (Figure 5(aii)). First, we measured the effect of retaining the strain at 0.1% for 100 s before increasing it to 100% for 100 s. This process was performed for a total of five shear/recovery cycles. Notably, at 0.1% strain, the hydrogel preserved its elastic behavior (G′ > G″), while 100% strain results in a full loss of the viscoelastic properties. However, after returning to a 0.1% strain, the broken PGB gel returned to its initial elastic state almost with a ~90% immediate recovery in all five cycles. After 100 s at 0.1%, even full viscosity was achieved. Noteworthy, a slight decrease in G′ values was observed during the shear/recovery cycles. These results are consistent with the concept that when the hydrogel experiences prolonged deformation cycles, it gradually loses its elastic properties due to an increasingly fluid-like behavior [69,70,71]. This outstanding self-healing ability of these hydrogels may be explained by their dynamic and adaptive non-covalent interaction network, which can rapidly reform following stress signals and thus reestablish the hydrogel network without the need for external factors. Due to these highly favorable thixotropic properties, PGB formulations may be suitable for injectable soft hydrogel applications.

We also performed a peak hold experiment (Figure 5(aiii)) in which the hydrogel is subjected to a brief shearing stress followed by a recovery time while the viscosity changes are recorded. For the first 10 s, an initial shear rate of 3.45 s^−1^ was applied, corresponding to the expected shear rate during 3D printing. Then, the rate was lowered to 0.1 s^−1^ to allow for hydrogel recovery. Importantly, following an initial decrease in viscosity likely caused by flow instability and polymer disentanglement, the viscosity stabilized after ~3 s of low shear stress [72]. This confirms the dynamic nature and highly adaptive non-covalent interaction network of the PGB hydrogel, as observed also in our dynamic step-strain experiment and demonstrates that the developed PGB ink is suitable for multi-layer 3D printing as the extruded filaments rapidly stabilize [73].

Overall, our results demonstrate that the developed PGB hydrogel has strong thixotropic and self-healing properties that are comparable to GB hydrogels (Appendix A), making it a viable choice for 3D-printing applications.

### 2.5. Morphological Analysis of PGB Hydrogel

The nanofibrillar structure of PGB hydrogels is highlighted in Figure 5b as an insert via scanning electron microscopy (SEM) cross-section analysis. Such a network provides the essential structural support for cell growth and proliferation [74]. The diameters of the formed nanofibers were also determined, with PGB hydrogels having a range from 5.5 to 30 nm and an average diameter of 14.02 ± 0.13 nm, while GB hydrogel had a range from 4 to 40 nm and a diameter average of 13.90 ± 0.53 nm (Appendix A). Additionally, the size distribution showed a full width at half maximum (FWHM) of 9.36 ± 0.49 and 12.03 ± 1.49 for PGB and GB hydrogels, respectively. Importantly, the determined nanofiber parameters are well within the range of natural ECM [75] and thus well suited to facilitate cell growth and proliferation.

### 2.6. Antimicrobial Properties of the PGB Hydrogels

Following a thorough characterization of the PGB hydrogels, we investigated whether they possessed antibacterial properties due to the incorporation of PDA. Thus, we studied bacterial adhesion as well as early biofilm formation of *E. coli*, *S. aureus*, and *S. epidermidis* on such scaffolds. These three strains were chosen as they have been linked to infections after biomaterial implantation. Notably, the majority of implant-related infections are caused by the *Staphylococcus* species, with *S. aureus*, a Gram-positive bacterium, being the most commonly isolated organism present in the early stages of infections, while *S. epidermidis* is found in the majority of later infection stages [76]. In addition, the Gram-negative bacterium *E. coli* was tested as it is one of the most common organisms associated with BAIs overall [77]. Of note, while *S. aureus* thrives at slightly neutral to slightly alkaline pH (7.2–7.6), *E. coli* (6.5-7.0) and *S. epidermidis* (5.0–7.0) prefer a neutral to slightly acidic pH. However, they are all relatively insensitive to pH changes around neutral pH [78,79]. As our hydrogels were designed for physiological pH (~7.4) to enable host cell infiltration, proliferation, and differentiation, the environment is equally well suited for antimicrobial testing, during which we did not observe any impact on hydrogel stability.

#### 2.6.1. Bacterial Adhesion on PGB Hydrogels

Bacterial adhesion is commonly thought to be mediated by van der Waals interactions, in which bacteria need to overcome energy barriers such as electrostatic repulsion to reach the biomaterial surface and eventually form surface-attached colonies via reversible and irreversible adhesion [80]. Thus, bacterial adhesion experiments were performed after 2 h incubation of the PGB and GB hydrogels with *E. coli*, *S. aureus*, and *S. epidermidis* (Figure 6a). As hypothesized, PGB hydrogel decreased the adhesion of all three bacteria strains, albeit with varying degrees of efficacy across strains. For example, while *S. epidermidis* showed the greatest decrease in adhesion, *E. coli* showed only a statistically insignificant reduction. This, however, is consistent with the widely reported fact that bacterial strains have different susceptibility profiles to different antibiotic treatments [81,82].

To further evaluate the antibacterial properties of PGB hydrogels, the bacterial live/dead ratio was determined (Table 1). Importantly, PGB hydrogels immersed in the respective bacterial suspensions exhibited a higher ratio (i.e., more dead cells) than the control hydrogel lacking PDA. After 2 h of incubation, this effect was most prominent for *S. epidermidis*, where a ratio of 0.10 vs. 0.83 was observed for GB and PGB hydrogels, respectively, indicating eight times more bacterial killing due to the incorporated PDA. While the effect was less dramatic for *E. coli* and *S. aureus*, the number of dead bacteria was still more than double for *E. coli* (0.16 vs. 0.38) and more than triple for *S. aureus* (0.08 vs. 0.29) when PDA was incorporated. Thus, the observed antibacterial properties can be predominantly attributed to the incorporation of the PDA.

Images of viable (green dots) and dead bacterial cells (red dots) obtained by confocal laser scanning microscopy (CLSM) are shown in Figure 6b. Due to the incorporation of PDA, PGB hydrogels show a reduced number of viable bacteria (i.e., less green signal) compared to GB hydrogels, while the number of dead cells is increased (i.e., more red signal).

Both quantitative and qualitative assays demonstrated that the incorporation of PDA in the GB hydrogel decreased bacterial adhesion; however, the efficacy varied depending on the strain tested. The strongest antibacterial properties were found against *S. epidermidis*, followed by the likewise Gram-positive bacterium *S. aureus* and the Gram-negative *E. coli*. Importantly, such variations are common in the antibacterial research field and are typically attributed to differences in overall cell wall structure (e.g., Gram-negative vs. Gram-positive bacteria) as well as variability in cell wall composition and thickness across individual strains [81].

#### 2.6.2. Evaluation of Biofilm Formation on PGB Hydrogels

In addition, the antibacterial effect on early biofilm formation was evaluated (Figure 6a,b, and Table 1). After 24 h, a statistically significant reduction in early biofilm formation was detected against all three strains. Notably, when the percentage of inhibition is considered, the anti-biofilm effectiveness against all three strains coincides well with the results of the bacterial adhesion experiments.

### 2.7. Cell Experiments

rMSCs were seeded drop-by-drop onto the PGB hydrogels and left for cell attachment at 37 °C and 5% CO_2_ for 4 h. Importantly, every 30 min, drops of fresh pre-warmed medium were added onto the scaffold to prevent cells from drying out.

#### 2.7.1. Cell Viability and Proliferation Experiment

We first evaluated the biocompatibility of our GB and PGB hydrogels by cell viability and LDH proliferation assays where rMSCs were incorporated into the printed hydrogels, and the cell viability was monitored for up to 14 days. As shown in Figure 7(ai), rMSCs consistently increased in quantity regardless of the hydrogel employed. Notably, a significant increase in the number of cells was observed for the PGB hydrogel when compared to the control GB hydrogel. However, the apparent proliferation rate was slightly lower than on tissue culture polystyrene (TCPS) plates alone. Next, the cell viability was evaluated relative to TCPS plates. Notably, while relative cell survival on PGB hydrogels was ~90% on days 1 and 3, it increased to ~110% on days 7 and 14 (Figure 7(aii,b)). GB hydrogels, on the contrary, started with a cell viability of ~90% on days 1 and 3, which dropped to ~75% on days 7 and 14. These results confirm previous reports that the incorporation of PDA has no cytotoxic side effects and, in fact, promotes cell survival [27,28,31]. This might be explained by the capacity of PDA-coated surfaces to interact with cell-adhesive proteins from the plasma membrane or other cell surface structures, thus promoting cell adhesion and proliferation, or by their ability to absorb proteins from the medium or the surrounding environment, better mimicking biological surfaces [83,84]. Taken together, these experiments confirmed that our developed PGB hydrogels provide a suitable environment for cell survival and proliferation for at least 14 days.

#### 2.7.2. Cell Morphology Experiment

The cell morphology of rMSCs was evaluated using CLSM and staining of nuclei and actin filaments. Figure 7(bii) demonstrates the successful adhesion of rMSCs onto the hydrogel after 14 days. However, while cells seeded onto TCPS plates exhibited a well-defined cytoskeleton, only an initial cytoskeleton definition and a spread-roundish cell morphology could be detected for our PGB hydrogels. However, while the observed cell morphology may be imperfect, both cell proliferation and survival confirmed that our PGB hydrogels can be functionalized with living cells, and the introduction of additional bioactive molecules may improve cell differentiation in the future.

#### 2.7.3. Cell Migration Experiment

A common problem of cell-functionalized scaffolds is the often poor colonization of the scaffold core [85,86]. This might be caused by excessive cell growth on the scaffold surface, leading to cell accumulation and, as a result, sealing of the scaffold’s pore structure and halting cell migration and/or nutrition uptake into the scaffold. Thus, we studied cell migration into the PGB hydrogel by analyzing the z-projection of the cells’ location on days 1, 3, and 7 (Figure 8). While the majority of rMSCs were found on the surface on day 1, the first cells had penetrated the scaffold by day 3. Over the following days, cells dispersed along the nanofibrillar hydrogel network, and by day 7, cells had colonized the hydrogel core, and a fair dispersion of cells within the scaffold was observed. Additionally, we calculated the migratory speed of rMSCs in z direction at these time points and determined values of 9.0 ± 4.6, 5.1 ± 0.2, and 2.5 ± 0.3 µm h^−1^, respectively. The quickest z-migration was found on day 1, which might be explained by the fact that over time, cells colonize the scaffold not only in z (penetration) but also in x and y (dispersion) directions, and the latter are not detected in our experimental set-up. Notably, recent studies have suggested that PDA coatings may boost migratory speed due to fibronectin depositions and subsequent integrin activation [87,88]. Likewise, the presence of PDA may modify the focal adhesion features of the hydrogel and thus facilitate improved cell movement.

#### 2.7.4. Alkaline Phosphatase (ALP) Activity

One reason to incorporate PDA into the hydrogel was to increase the osteogenic activity of the GB hydrogels since several reports have demonstrated its ability to stimulate osteogenic differentiation in vitro [27,28,88]. Thus, we tested for alkaline phosphatase (ALP) activity as it is a strong indicator for early-stage osteogenesis (Figure 9). Notably, a significant increase in ALP activity was detected after 7 days of cell culture, both compared to GB hydrogels and rMSCs cultured alone, followed by a decrease at 14 days to levels seen after 3 days. Importantly, this is not surprising as PDA mainly plays a role in early cell differentiation [63]. These results may indicate that the presence of PDA may be sufficient to elicit an osteogenic response of rMSCs in GB hydrogels; however, additional biomolecules may improve differentiation. Nevertheless, the detected ALP expression and activity clearly demonstrate the functionality of the hydrogel-embedded rMSCs and prove that our PGB hydrogel scaffold can indeed be functionalized by living cells.

## 3. Materials and Methods

### 3.1. Materials

Guanosine (Guo), dopamine hydrochloride (DA), boric acid (BA), phosphate-buffered saline (PBS), glutaraldehyde, hexamethyldisilazane (HDMS), bovine serum albumin (BSA), penicillin-streptomycin (Pen-Strep, P/S), research-grade fetal bovine serum (FBS), Accutase cell detachment solution, paraformaldehyde (PFA), Triton X-100, fluorescein isothiocyanate-dextran (FITC-Dextran; MW of 70, 500, and 2000 kDa), alkaline phosphatase yellow liquid substrate system for ELISA, p-nitrophenol solutions, tryptic soy broth (TSB), Phalloidin-Atto 488, and DAPI ready-made solution were all purchased from Sigma-Aldrich (Saint Louis, MO, USA). Potassium hydroxide (KOH) pellets EMPLURA^®^ were purchased from Merck KGaA (Darmstadt, Germany). Advanced DMEM (Dulbecco’s Modified Eagle Medium), GlutaMAX™ Supplement sterile syringe filters (polyethersulfone, 0.2 µm, 25 mm), ReadyProbes™ Cell Viability Imaging Kit (Blue/Green), and CellTracker™ Deep Red were purchased from Thermo Fisher Scientific (Waltham, MA, USA). Invitrogen™ CyQUANT™ LDH Cytotoxicity Assay, Invitrogen™ LIVE/DEAD™ BacLight™ Bacterial Viability Kit, and Difco agar bacteriological were purchased from Thermo Fisher Scientific (Waltham, MA, USA). Luria broth (LB, Miller’s LB broth) was purchased from Condalab (Torrejón de Ardoz, Madrid, Spain).

All solutions were prepared with ultrapure water from a Milli-Q Gradient A10 water purification system (EMD Millipore, USA) with a resistance ≥18 MΩ cm at 25 °C and a total organic carbon (TOC) count of ≤4 ppb.

### 3.2. Optimization and Assessment of PGB Hydrogel Formation

PGB hydrogels were fabricated by the addition of accurately measured amounts of DA (final concentrations 2.5, 5.0, 7.5, 10, and 12.5 mM) in a 4 mL glass vial with 533 µL of 300 mM BA (40 mM final), 533 µL of 300 mM KOH (40 mM final), and 2933 µL of Milli-Q water in a total volume of 4 mL. The obtained mixtures were heated to 80 °C under continuous magnetic stirring for 30 min, even though the solutions became translucent already after approx. 5 min. After the incubation time and without disturbance, the mixture was transferred to a new 4 mL glass vial containing 102 mg of Guo (90 mM final). The mixture was maintained at 80 °C with stirring, and after approx. 15 min, the solutions became translucent again. Hydrogel formation was observed when the solutions were cooled to room temperature (RT) over 5 h to ensure complete gelation and polymerization of DA into PDA. The hydrogelation of the mixtures was assessed by inversion test (i.e., turning the tubes upside down). Three samples were studied for each condition.

The amount of PDA polymerized in the guanosine-based hydrogels was assessed by fluorescence intensity measurements. In brief, 500 µL of the heated and translucent PGB solution was transferred into clean 1.5 mL microcentrifuge tubes (Eppendorf AG, Hamburg, Germany) and incubated at RT for different time spans (0, 2, 4, 8, and 24 h), after which they were placed at −80 °C. Then, the samples were lyophilized using a laboratory freeze dryer (LyoMicron, Coolvacuum Technologies, S.L., Barcelona, Spain) for 48 h with an ice condenser temperature of −55 °C. Lyophilized samples were then mixed with 500 µL of Milli-Q water and maintained at 50 °C in a Thermo-Shaker PHMT-PSC18 (Grant Instruments, Cambridgeshire, UK) until fluorescence measurements to avoid hydrogelation. To correlate measured fluorescence values to the amount of polymerized and hydrogel-embedded PDA, a calibration curve was prepared using PDA, and the fluorescence intensity was measured at 310 nm after excitation at 260 nm using a microplate reader (Infinite M Nano, TECAN, Switzerland) set to 40 °C. Additionally, DA, Guo, KOH, and Milli-Q water fluorescence scans were recorded from 290 to 700 nm with the exact same excitation wavelength (260 nm) to confirm that there are no interferences in the readings (Appendix A). Three samples were studied for each condition.

### 3.3. Semi-Quantification of Printability

PGB hydrogels were prepared as described above. After the mixture was heated up to 80 °C with stirring for 15 min, the translucent solution was transferred into a 3 mL syringe with a parafilm-sealed tip, and the mixtures were allowed to cool down without disturbance at RT for 5 h. Then, the jellified samples were printed with a Cellink BIO X^TM^ bioprinter (Cellink, Gothenburg, Sweden) at 37 °C and a 20-gauge nozzle (Cellink, Gothenburg, Sweden) with an inner diameter of 0.58 mm. The printing conditions were as follows: nozzle and substrate table temperature at 37 and 25 °C, respectively, and a printing speed of 2 mm s^−1^. Images of the printed hydrogels were analyzed using ImageJ [57] by measuring (i) the area under the printed and free-floating filaments, (ii) the perimeter of the printed squares, and (iii) the area and angle between interconnected filaments (n = 3) (Figure 10). Three samples were studied for each condition and experiment.

#### 3.3.1. Filament Collapse Test

The filament collapse test was performed as described elsewhere [54,55]. In short, a platform consisting of seven pillars spaced in predefined intervals of 1, 2, 3, 4, 5, and 6 mm was designed within the solid modeling CAD software SolidWorks (Dassault Systèmes, Waltham, MA, USA) and printed using polylactic acid (PLA) as material and an Ultimaker 2+ 3D printer (Utrecht, The Netherlands). The dimensions of the five central pillars and the two corner pillars were 2 × 10 × 6 mm^3^ (W, H, D) and 5 × 10 × 6 mm^3^, respectively.

The C_f_ (Equation (1)) of the printed hydrogel was determined by the mid-span deflection of the hanging filament. After 5 h of hydrogelation and polymerization in the syringe at room temperature, a filament was deposited onto the platform (Figure 10a) with the nozzle tip being placed 0.3 mm above the top surface of the pillars, and printing was stopped 5 mm after the last pillar. An image of the deposited filament was taken 1 min after the printing and the area under the filament was calculated using ImageJ [57]. The C_f_ is calculated as the percentage of the deflected area after filament suspension versus the theoretical area [54,55]:(1)$$C_{f}=\frac{A_{t}^{c}-A_{e}^{c}}{A_{t}^{c}}\times 100,$$
where $A_{e}^{c}$ and $A_{t}^{c}$ are the experimental and theoretical area under the filament, respectively.

#### 3.3.2. Filament Fusion Test

Ideal hydrogels exhibit a smooth surface and a constant width throughout the printed filament, allowing the generation of regular grids and squares in the printed 3D system. The filament fusion test provides crucial insights into the grid and hole morphology of the printed network. A pattern with increasing filament-to-filament distance from 1 to 5 mm in 1 mm increments was printed (Figure 10b), and a picture of the generated scaffolds was taken after 1 min of settling time. Three key parameters were determined: (i) the D_fr_, which expresses the rate of material spreading in percent; (ii) the P_r_, which expresses the shape of the printed holes; and (iii), the D_a_, which describes the rate of non-rectangularity of the printed grid squares. Importantly, while low values for material spreading and angular deviation are ideal, the printability score should be close to 1. Of note, the angle deviation calculation was performed for the 5 × 5 mm^2^ square, and the equations used are detailed below [54,55]:(2)$${\mathrm{Df}}_{r}=\frac{A_{t}^{f}-A_{e}^{f}}{A_{t}^{f}}\times 100$$
(3)$$P_{r}=\frac{L^{2}}{16A_{e}^{f}}$$
(4)$$D_{a}=\frac{\theta_{t}-\theta_{e}}{\theta_{t}}\times 100$$
where $A_{t}^{f}$ and $A_{e}^{f}$ represent the theoretical and experimental area of the pore, respectively, L is the perimeter of the pore, and θ_t_ and θ_e_ are the theoretical and experimental angles, respectively.

#### 3.3.3. Printability Score

To enable a comparison between the various hydrogel compositions, a semi-quantitative scoring scheme (Appendix A) was developed and points assigned for each measured parameter.

For hydrogels with a little or no filament deflection in the filament collapse test (C_f_ between 0 and 5%), a maximum score of 40 was assigned. Likewise, the filament fusion test has a maximum score of 40 for all three parameters determined. Hydrogels without any material spreading and at a perfect 90° angle exhibit a D_fr_ and D_a_ of 0% and thus also get 40 points, while printability scores should be close to 1, equivalent to grid holes with a square shape. For smaller values of P_r_ and larger values of D_fr_ and D_a_, a smaller degree of hydrogel gelation is attributed, and a lower score is attributed. In order to allow an overall comparison, the obtained scores of each hydrogel formulation were summed and normalized to a scale from 1 to 10, with 10 being the best shape fidelity.

### 3.4. Degradation of PGB Hydrogels

Three-dimensionally printed PGB hydrogels were immersed in fresh Advanced DMEM complete medium, and the hydrogels were monitored for up to 14 days at 37 °C in a humidified incubator and 5% (*v*/*v*) CO_2_. The medium was exchanged every 2 days, and the stability of the hydrogel was documented at distinct time points (0, 1, 3, 5, 7, and 14 days) by taking pictures of the printed material and the pore scaffold area calculated in ImageJ [57]. Three samples were studied for each condition.

### 3.5. Permeability and pH Studies of PGB Hydrogels

The permeability of PGB hydrogels for nutrients and small molecules was determined by using FITC-Dextran variants with a molecular weight of 70, 500, and 2000 kDa [89]. PGB hydrogels were printed in a 6-well plate, immersed in 4 mL of FITC-Dextran solution (1 mg mL^−1^ in complete Advanced DMEM medium), and incubated at 37 °C for 10 h. Then, 100 µL of the supernatant was transferred into a 96-well plate, and the fluorescence intensity (λ_ex_ = 450 nm, λ_em_ = 550 nm) was measured using a multimode plate reader (Infinite M Nano, TECAN, Switzerland). Scaffolds were rinsed three times with PBS to remove residual FITC-Dextran, fresh medium was added, and incubation continued for another 10 h. Then, 100 µL of the medium was transferred again into a 96-well plate, and the fluorescence intensity was measured. This procedure was repeated twice, and three replicates were analyzed for each FITC-Dextran solution and time point. Moreover, the pH of the GB and PGB hydrogels was determined using a Mettler Toledo pH meter (Greifensee, Switzerland).

### 3.6. Rheological Studies of PGB Hydrogel

The PGB hydrogel solution was formed by mixing DA, BA, KOH, and Guo (5 mM, 40 mM, 40 mM, and 90 mM, respectively) as described in section “Optimization and Assessment of PGB Hydrogel Formation”. Polylactic acid (PLA) molds (20 mm diameter) were designed by using SolidWorks software (Dassault Systèmes, Waltham, MA, USA) and printed with a Sigma R19 3D printer (BCN3D technologies, Gavà, Barcelona, Spain). A total of 2 mL of PGB hydrogel solution was transferred into the PLA molds, and after cooling down for 1 h, the formed hydrogel was gently transferred from the mold and the rheological properties were assessed using a Discovery HR-2 hybrid rheometer (TA Instruments, New Castle, DE, USA). All rheology measurements were made using two different 20 mm Peltier geometries, and the samples were placed carefully onto the surface of the lower plate. For the oscillatory tests, an amplitude strain sweep with a parallel stainless Peltier was carried out at a frequency of 1 Hz and deformation ranging from 0.01 to 100%. The subsequent assays were performed using controlled deformation values which were fixed based on the range of linear viscoelasticity (LVE) determined through the amplitude sweep measurements. In the 10-cycle dynamic step-strain sweep test, a cross-hatched Peltier plate was used, and the applied strain value was cycled between 0.1 and 100% for a total of 100 s at a constant angular frequency of 10 rad s^−1^. Both the elastic properties and fluidity were measured by the G′ and the G″, which were monitored during strain sweep, dynamic step-strain sweep, and peak hold assay. All experiments were repeated at least 5 times.

### 3.7. Morphological Analysis of PGB Hydrogel

A PGB hydrogel was prepared by mixing DA (5 mM), BA (40 mM), KOH (40 mM), and Guo (90 mM), 3D printed (as detailed in section “Optimization and Assessment of PGB Hydrogel Formation”), and immersed in a fixative solution of 1.0% (*w*/*v*) glutaraldehyde in PBS for 1 h at RT. The stabilized scaffolds were then washed three times in phosphate-buffered saline (PBS) and dehydrated using a series of ethanol washes (25%; 50%; 70%; three times 85%; three times 95%; and three times 100%), each for 10 min. Next, samples were immersed in hexamethyldisilazane (HDMS) for 10 min, vacuum dried at RT, and coated with gold (Sputter coater AGB7340, Agar scientific, Stansted, UK) for SEM (Zeiss Merlin FE-SEM microscope, Carl Zeiss NTS GmbH, Germany) analysis. Three samples were studied, and five images were taken at different Z-levels, a working distance of 5 mm, and a potential of 2 kV.

### 3.8. Antimicrobial Properties of 3D-Printed PGB Hydrogels

Bacterial assays were performed with three different oral bacteria: *E. coli*, *S. aureus*, and *S. epidermidis.* While *E. coli* was grown on Luria broth (LB), *S. aureus* and *S. epidermidis* were maintained on tryptic soy broth (TSB). Solution cultures were freshly incubated from a single colony overnight at 37 °C before each assay. The optical density of each bacterial suspension was adjusted to 0.1 ± 0.01 at 600 nm, giving approximately 0.8 × 10^8^ colony-forming units (CFUs) per mL of culture. The assays were performed in static conditions due to the short testing time. All assays were performed using three replicates for each condition.

#### 3.8.1. Bacterial Cell Adhesion to PGB Hydrogels

PGB and GB hydrogels were placed into 6-well plates and each incubated with 4 mL of bacterial suspension (1 × 10^8^ CFU mL^−1^) for 2 h at 37 °C. After this time, the medium was aspired, and the samples were gently washed twice with PBS. PGB hydrogels were then transferred into new 6-well plates, and hydrogel-adhered bacteria were detached by vigorous shaking and solubilizing the hydrogel in 2 mL of PBS. The obtained cell solutions were seeded in serial dilutions on LB and TSB agar plates for *E. coli* and *Staphylococcus*, respectively. The plates were then incubated at 37 °C for 24 h, and the CFUs were counted. Three samples were studied for each condition.

#### 3.8.2. Evaluation of Early Biofilm Formation on PGB Hydrogels

PGB and GB hydrogels were placed into 6-well plates and incubated with 4 mL of bacterial suspensions (1 × 10^8^ CFU mL^−1^) for 2 h at 37 °C. Then, PGB hydrogels were washed twice with medium to remove non-attached cells, and attached bacteria were allowed to grow for 24 h at 37 °C surrounded by 4 mL of fresh medium. Then, PGB hydrogels were washed again 3 times with PBS, and the PGB hydrogels were transferred into 15 mL falcon tubes. To break the hydrogels, 1 mL of PBS was added by thoroughly pipetting up and down and gentle vortexing. To determine the number of attached bacteria after 24 h of growth (corresponding to approx. 35–50 doubling times), these bacteria were then seeded using serial dilutions on LB plates for *E. coli* and on TSB agar plates for *S. aureus* and *S. epidermidis*. The plates were incubated at 37 °C, and the CFUs were counted after 24 h. Three samples were studied for each condition.

#### 3.8.3. Viability of Bacteria on PGB Hydrogels

A LIVE/DEAD BacLight Bacterial Viability Kit was used to determine the viability of the bacteria adherent to the PGB hydrogels. The green-fluorescent nucleic acid stain SYTO^®^ 9 (485/530) was used to label all bacterial cells as it can penetrate cells with both intact and damaged membranes. On the contrary, the red-fluorescence nucleic acid stain propidium iodide (485/630) can only penetrate cells with damaged cell membranes and was thus used to label dead bacteria. Using the appropriate dye mixture, cells with intact cell membranes were stained fluorescent green (living cells) while damaged bacteria were stained also fluorescent red. Three independent experimental repeats and technical triplicates were analyzed.

Bacterial suspensions were incubated with PGB hydrogels as specified in sections “Bacterial Cell Adhesion” and “Evaluation of Early Biofilm Formation on PGB Hydrogels”, and after certain incubation times, the supernatant was removed and hydrogels were washed three times with PBS. Then, the hydrogels were incubated with 2 mL of dye-containing solution (1.5 µL of SYTO^®^ 9 and 1.5 µL of propidium iodide in 1 mL of 0.85% NaCl solution (0.14 M; isotonic solution)) at RT for 15 min in the dark. CLSM images of the attached bacteria were acquired using a Leica TCS SP8 confocal laser scanning microscope (Leica Microsystems GmbH) equipped with an argon laser and using excitation/emission wavelengths of 488/500–560 and 488/600–650 nm, respectively. ImageJ [57] software was used to count bacteria, and at least three independent experimental repeats and technical triplicates were analyzed. The ratio of red fluorescence (dead cells) vs. green and red fluorescence (dead and live cells) was calculated as the portion of killed cells for each treatment.
(5)$$\mathrm{Volume}\;\mathrm{ratio}\;\mathrm{of}\;\mathrm{dead}\;\mathrm{cells}=\frac{\mathrm{volume}\;\mathrm{of}\;\mathrm{red}\;\mathrm{bacteria}}{\mathrm{volume}\;\mathrm{of}\;\mathrm{red}\;\mathrm{bacteria}+\mathrm{volume}\;\mathrm{of}\;\mathrm{green}\;\mathrm{bacteria}\;},$$

### 3.9. Cell Experiments

#### 3.9.1. rMSC Isolation and Culture

The rMSCs were isolated from bone marrow flushed from the femurs and tibias of two five-weeks-old male Sprague-Dawley rats. In short, rats were euthanized by CO_2_ inhalation for the collection of tibias and femoral bones from the hind legs, and cells were harvested, passed through a 40 µm cell strainer, and centrifuged for 3 min at 200 rcf. Cells were cultured in 75 cm^2^ culture flasks at 37 °C and 5% CO_2_ in culture medium consisting of Advanced DMEM medium supplemented with glucose (4500 mg L^−1^), sodium pyruvate (110 mg L^−1^), 20% (*v/v*) fetal bovine serum (FBS), 1% (*v/v*) GlutaMAX, and 1% (*v/v*) penicillin/streptomycin (10,000 U mL^−1^ and 10 mg mL^−1^, respectively).

The medium was changed every 2–3 days, and subconfluent cells were detached from the culture flask by adding 4 mL of Accutase solution and 5 min of incubation. Only cells from passages 2 to 5 were used for experiments, and all cell-based assays were performed as three independent biological experiments with three technical replicates each.

#### 3.9.2. rMSC Seeding

For synthesis of sterile PGB hydrogels, the required amounts of Guo were sterilized under ultraviolet (UV) light for 15 min, whereas aqueous stock solutions of BA, KOH, and Milli-Q water were filtered using 0.2 µm polyethersulfone (PES) syringe filters as described in experimental section “Semi-Quantification of Printability”. Magnets and nozzles were sterilized by submersion in 70% ethanol for 30 min followed by three washes with sterile PBS. After 5 h of hydrogel maturation, 6-layer scaffolds were printed by filament extrusion using a BioX 3D printer in 6-well plates followed by drop-to-drop addition of rMSCs suspension (3.75 × 10^6^ cells mL^−1^ in drops of 5 µL with a final amount of 200 µL) on top of the printed scaffold. To prevent cells from drying out, every 30 min, a fresh drop of complete medium was added onto the scaffold surface attached to the hydrogel scaffold. All assays were performed as three biological and technical replicates.

##### 3.9.3. Cell Viability and Proliferation Experiment

rMSCs seeded onto PGB hydrogels were immersed in complete medium and incubated at 37 °C and 5% CO_2_ for 1, 3, 7, and 14 days. After incubation, medium was carefully removed from the respective well, and samples were rinsed three times with PBS. For the cell viability assay, the medium was replaced with fresh medium (4 mL) together with 8 drops each of the NucBlue^®^ Live (Hoechst 33342) and NucGreen^®^ Dead reagent (ReadyProbes™ Cell Viability Imaging Kit) and incubated for 30 min at 37 °C and 5% CO_2_. Then, samples were washed three times with PBS and imaged with a Leica TCS SP8 CLSM equipped with an argon laser and using excitation/emission wavelengths of 405/440–480 and 488/503–543 nm, respectively. ImageJ [57] software was used for cell counting, and at least three independent experimental repeats of technical triplicates were analyzed.

Cell proliferation was assessed by detecting lactate dehydrogenase (LDH) activity in the cell culture medium after induced cell lysis using the commercially available Invitrogen™ CyQUANT™ LDH Cytotoxicity Assay. This assay is based on the conversion of lactate to pyruvate in the presence of LDH concurrent with the reduction of NAD^+^ to NADH. After certain incubation times, PGB hydrogels with rMSCs seeded on top were washed 3× with PBS and subsequently treated with 200 µL of 0.2% Triton X-100 in PBS to destroy the hydrogel and lyse the cells. Then, 50 µL of the cell culture media was mixed with 50 µL assay reagent in a 96-well plate and incubated for 30 min at RT in the dark. Finally, absorbance at 490 and 680 nm was recorded using a microplate spectrophotometer system (Infinite M Nano, TECAN, Switzerland). All assays were performed as three biological and technical replicates.

##### 3.9.4. Cell Morphology Experiment

Cell morphology was determined at days 1, 3, and 7 by staining the actin filaments and the nuclei of rMSCs in PGB hydrogels. Controls included cells alone and GB hydrogels. To do so, rMSCs-PGB hydrogels were incubated for different time spans in complete medium at 37 °C and 5% CO_2_. After incubation, scaffolds were washed twice in PBS to remove loosely attached cells, followed by fixation in 4% paraformaldehyde (PFA, in PBS) for 30 min and three additional PBS washes. Then, cells were permeabilized by submerging the scaffolds in PBS-T (0.1% Triton X-100 in PBS) for 20 min at RT, followed by immersion in a 1% bovine serum albumin (BSA) in PBS for 1 h to minimize non-specific background binding of the dye. Next, rMSCs-PGB hydrogels were submerged in a solution of phalloidin-Atto 488 (0.1 µg mL^−1^; in PBS) for 1 h at RT in the dark to stain actin filaments, followed by three washes with PBS. Then, the hydrogels were incubated again in the dark for 5 min at RT in a solution of DAPI (20 µg mL^−1^; in PBS), and after additional three washes with PBS, cells embedded in the hydrogel were imaged using a CLSM equipped with an argon laser, with excitation/emission wavelengths set to 405/440–480 and 488/495–545 for DAPI and phalloidin-Atto 488 detection. A 63× oil immersion objective was employed for imaging. Three samples were studied for each condition.

##### 3.9.5. Cell Migration Experiment

Before seeding the cells onto 3D-printed PGB hydrogel scaffolds, adherent rMSCs were treated with 5 μM CellTracker™ Deep Red in serum-free medium and incubated at 37 °C for 30 min, as described in the manufacturer’s protocol. Then, rMSCs were seeded on top of 3D-printed PGB hydrogels, which were prepared as described above (experimental section “rMSC Seeding”). Migration of Deep Red labeled cells through the hydrogel was monitored at days 1, 3, 7, and 20 using bright field z-stack images over a slide of 500 µm and using a CLSM (Leica Microsystems GmbH) equipped with an argon laser and a 10× objective with excitation/emission wavelengths set to 630–660 nm. Cell migration speed (CMS) was determined by tracking the z-displacement of the labeled cells and following the formula
(6)$$\mathrm{CMS}=\frac{d}{t},$$
where d represents the distance and t the time. All assays were performed as three biological and technical replicates.

##### 3.9.6. Alkaline Phosphatase (ALP) Activity

The bioactivity of cell-functionalized PGB hydrogels was assessed by quantifying ALP activity of rMSCs at days 1, 3, 7, and 14 after cell seeding. For that, the same procedure as for the cell proliferation was followed (see experimental section “Cell Viability and Proliferation Experiment ”). At distinct incubation times, printed scaffolds with attached cells were rinsed 3× with PBS, and cells were lysed using 200 µL of 0.2% Triton X-100 in PBS, which also destroyed the hydrogel scaffold. Then, supernatants were collected and incubated with p-nitrophenol phosphate (pNPP). Note: when it was not possible to perform the ALP activity assay right after cell lysis, samples were collected and stored at −80 °C. Next, solutions were incubated at RT in the dark for 30 min, and a calibration curve was prepared using p-nitrophenol (p-nitrophenol in 0.2% Triton X-100) as a standard. ALP activity was calculated as pg of reaction product (p-nitrophenol) per min (over 30 min in total) and normalized to the number of cells obtained in the corresponding cell proliferation assay.

### 3.10. Statistical Analysis

A non-parametric Mann–Whitney U-test was used to determine the significance of any differences between two groups. A non-parametric Kruskal–Wallis test was used to analyze significant differences among groups. The significance level was set at *p* < 0.05.

## 4. Conclusions

In this study, we synthesized and extensively characterized a novel PDA-augmented guanosine-based (PGB) hydrogel that outperforms traditional hydrogels in terms of stability, antibacterial activity, and biological properties. To maximize PDA impact, we first optimized PDA incorporation and found that 5 mg mL^−1^ of DA and an incubation time of 5 h were optimal. After assessing 3D printability, we used SEM and rheological studies to show that the PGB hydrogel possesses a strongly interconnected nanofibrillar network and good thixotropic properties. Moreover, as PDA is recognized as an antibacterial agent, we tested its effectiveness against *E. coli*, *S. aureus*, and *S. epidermidis* and found that our developed hydrogel significantly reduced bacterial adhesion and early biofilm formation against *Staphylococcus* when compared to standard GB hydrogels. Finally, we evaluated the hydrogel’s ability to sustain migration, proliferation, and differentiation of rMSCs seeded on top of the printed scaffolds. We found that rMSCs were able to penetrate and migrate within hydrogel scaffolds, and alkaline phosphatase activity analysis showed enhanced osteogenic differentiation. However, a spread but non-differentiated roundish cell morphology was observed with only early cytoskeleton definition, which we hope to improve by the future incorporation of additional bioactive compounds.

In conclusion, our novel PGB hydrogel has proven to be a highly promising scaffold for tissue engineering that can be easily functionalized post-printing with living cells and exhibits superior stability and antibacterial properties compared to traditional guanosine-based hydrogels. Furthermore, the developed formulation can be further enhanced by integrating additional bioactive molecules and moieties to improve cell attachment and differentiation or to elicit other cellular responses.

## Data Availability

The data presented in this study are available upon request from the corresponding author.

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
