# Peer review of "Polydopamine Incorporation Enhances Cell Differentiation and Antibacterial Properties of 3D-Printed Guanosine-Borate Hydrogels for Functional Tissue Regeneration"

_ijms, 2023, doi:10.3390/ijms24044224_

Round 1
Reviewer 1 Report
This research is interesting and worthy of publication. As a comment, the authors should introduce hydrogel-based tissue regeneration to claim this novelty by adding and comparing with these references.
Algainte doi.org/10.1016/j.actbio.2015.09.001
Gelatin doi.org/10.1016/j.reth.2021.11.006
Chitosan doi.org/10.1016/j.carbpol.2020.117023
Reviewer 2 Report
1. Scheme 1, is the phenylboronic acid lipid structure formed between the nano and the gel pH sensitive? How does this moiety change in the presence of Kimura and E. coli? What is the general pH of the microenvironment for S. aureus and E. coli infection?
2. Figure 5c, why does the viscoelasticity of the material increase instantaneously at 10 s instead of a slow process?
3. Where are these PDA-based hydrogels going to be used in real life? Advantages of the designed PDA-augmented guanosine-based hydrogel can be improved by comparing and citing 10.1016/j.ijbiomac.2020.06.283. The novelty of this work can be described at the end of the Introduction.
4. There are some formatting errors in the article. For example, spelling of references must be checked to meet the journal style (such as Reference 8). Please check carefully and use it properly.
5. The Conclusion section should be clear and concise: the important results and main conclusions drawn in this paper should be highlighted and presented in more precise language.
6. Section 2, clear information about the number of repetitions and statistical evaluation of the data is missing. This is crucial for high-quality research work.
Reviewer 3 Report
In this manuscript, the author reports, ‘Polydopamine incorporation enhances cell differentiation and antibacterial properties of 3D printed guanosine-borate hydrogels for functional tissue regeneration’. The authors should address the following questions before getting a possible publication.
Recommendation: Major revisions are needed as noted.
1. The novelty of the present article should be discussed a little bit more in the Introduction section.
2. The formatting and grammatical errors in the article need to be checked carefully.
3. The author should write the purpose for each test in one/two sentences (in brief) before explaining the results of the characterization techniques.
4. What does the error bar stands for in different Figures throughout the manuscript? It should be mentioned in Figure captions.
5. Abbreviations should be defined at their first instance such as rMSCs etc.
6. The authors are encouraged to include text in brief about the antibacterial mechanism of the PGB hydrogels.
7. The conclusion can be more precise.
8. The authors have cited relevant references in the Introduction section; however the manuscript needs to be highlighted with recent reports further to broaden the impact: Polymers, 13(21), 3782; Gels, 8(6), 356; ACS Applied Bio Materials 2022, 5, 5617–5633(https://doi.org/10.1021/acsabm.2c00664); Biomaterials, 230, 119598; Biomaterials Science, 8(22), 6190-6203; ACS Applied Materials & Interfaces, 12(46), 51940-51951
Round 2
Reviewer 3 Report
The authors have addressed all the questions raised before. Therefore the manuscript can be accepted in the present form
Author Response
We want to thank the reviewer once again for their time to check and assess our manuscript, and we are happy to hear that our revisions clarified all points raised.